# Enhancing Group Fairness in Federated Learning through Personalization

## Abstract

Personalized Federated Learning (FL) algorithms collaboratively train customized models for each client, enhancing the accuracy of the learned models on the client's local data (e.g., by clustering similar clients, by fine-tuning models locally, or by imposing regularization terms). In this paper, we investigate the impact of such personalization techniques on the *group fairness* of the learned models, and show that personalization can also lead to improved (local) fairness as an unintended benefit. We begin by illustrating these benefits of personalization through numerical experiments comparing several classes of personalized FL algorithms against a baseline FedAvg algorithm, elaborating on the reasons behind improved fairness using personalized FL, and then providing analytical support. Motivated by these, we then show how to build on this (unintended) fairness benefit, by further integrating a fairness metric into the cluster-selection procedure of clustering-based personalized FL algorithms, and improve the fairness-accuracy trade-off attainable through them. Specifically, we propose two new fairness-aware federated clustering algorithms, `Fair-FCA` and `Fair-FL+HC`, extending the existing `IFCA` and `FL+HC` algorithms, and demonstrate their ability to strike a (tuneable) balance between accuracy and fairness at the client level.

## 1 Introduction

Federated Learning (FL) has emerged as a pivotal paradigm for collaboratively training a shared machine learning model across distributed datasets/clients in a privacy-preserving manner (Kairouz et al., 2021). Shared, global models learned through FL can (potentially) outperform standalone models (those trained individually by each client in the absence of collaboration), owing to their ability to effectively aggregate knowledge from diverse clients. Popular FL algorithms such as FedAvg (McMahan et al., 2017) have been found to perform well, particularly when clients have independent and identically distributed (iid) datasets. However, real-world applications often involve heterogeneous datasets across clients, a situation where the performance of FedAvg can considerably deteriorate (Li et al., 2019b). Moreover, the standard FL approach yields a single global model, lacking customization for each client. Consequently, clients with heterogeneous datasets may experience low local accuracy (Li et al., 2020; Tan et al., 2022; Karimireddy et al., 2020). To address these limitations, a spectrum of *personalized FL* techniques has been proposed, to enhance the local accuracy of the learned models, while keeping some of the benefits of collaborative learning.

Aside from handling issues of data heterogeneity, training FL models that can uphold societal values, as formalized through notions of *algorithmic (group) fairness* (Barocas et al., 2019), has been a focus of recent research. Consider as an example the scenario where FL is used to train a foundation model/LLM on local datasets, contributed by diverse participants from different regions and communities (Kuang et al., 2023). Without careful oversight, the resulting model will favor language and content preferred by the majority contributors, often disregarding the unique linguistic nuances and cultural contexts of minority groups (Durmus et al., 2023). As another example, existing research (Kirdemir et al., 2021) finds structural and systemic bias in YouTube video recommendation systems. In this case, the collaboratively trained recommendation model begins to favor certain perspectives over others, inadvertently reinforcing biases present in the data. As a result, users are fed information that perpetuates stereotypes, causing harm to underrepresented communities. These examples motivate the development of FL algorithms that can meet certain fairness criteria.

In this paper, we establish an alignment between these two considerations in developing FL algorithms: *personalization can also improve fairness*. That is, we show that personalization techniques (which are designed to improve the local accuracy of the learned models) can also enhance algorithmic fairness as an unintended benefit (i.e., they can reduce the disparities in how the resulting model treats individuals from different protected groups, once the model is deployed by the client).

We begin by illustrating this alignment between personalization and fairness through numerical experiments comparing the average local fairness achieved by several personalized FL algorithms (IFCA (Ghosh et al., 2020), MAML-FL (Fallah et al., 2020), FedProx (Li et al., 2020), FL+HC (Briggs et al., 2020), and CFL-Cosine (Sattler et al., 2020)), against FedAvg (McMahan et al., 2017) and standalone learning (where each client learns by itself, and there is no collaboration), under three notions of fairness (Statistical Parity, Equality of Opportunity and Equalized Odds), and using both synthetic and real-world data ("Adult" dataset (Dua & Graff, 2017), and combinations of two different prediction tasks and two protected attributes in the "Retiring Adult" dataset (Ding et al., 2021)). Based on these experiments, we highlight two potential factors driving the alignment:

1. *Statistical* advantages of diverse data. Collaborative learning algorithms in essence have access to more "information" (data) from diverse clients. They can therefore enhance both accuracy and fairness, especially when dealing with imbalanced datasets, as in these scenarios issues of algorithmic unfairness can be attributed to the under-representation of samples from the disadvantaged group in the data. Further, personalized algorithms can outperform non-personalized collaborative learning (e.g. FedAvg), as (we posit) they are less prone to overfitting to the majority group's data.

2. *Computational* advantages due to alignments in local accuracy and fairness. Moreover, we identify cases where the sample distribution within clients is such that improving accuracy also promotes fairness. More intuitively, consider the concept of clustered FL, a personalized FL technique, where clients are organized into groups based on similarities in model performance (a proxy for data similarities). We argue that this clustering can be seen as a means to foster a model that is both more precise and more equitable, as it effectively treats information originating from other clusters as noise, which, if left unaddressed, would have led to model divergence. (Building on this intuition, we also provide analytical support under certain conditions in Propositions 1 and 2 in Appendix B).

To the best of our knowledge, this work is the first to identify the (unintended) fairness benefits of personalization in FL. Most existing works primarily focus on refining algorithms to attain either improved local accuracy (the personalized FL literature), *or*, enhanced (global) fairness (the fair FL literature); we discuss related work from each direction in Section 2. In contrast, our work examines the influence of existing personalization techniques on (local) fairness, and points to inherent features of collaborative learning and personalization that can advance fairness. Additionally, prior work points to the challenge of balancing the trade-off between fairness and accuracy of an algorithm, as enhancing fairness often comes at the cost of lower accuracy (Gu et al., 2022; Ezzeldin et al., 2023; Chien & Danks, 2023). In contrast, we identify instances that FL in general, and personalized FL in particular, can improve *both* accuracy and fairness compared to standalone learning.

Furthermore, inspired by our findings, we propose two new fairness-aware federated clustering algorithms, Fair-FCA and Fair-FL+HC, based on the existing IFCA (Ghosh et al., 2020) and FL+HC (Briggs et al., 2020) personalized FL algorithms, respectively. Both Fair-FCA and Fair-FL+HC algorithms modify the cluster identity assignment step to encompass a fairness criterion. Building on our finding that personalization can offer dual benefits, we demonstrate that this additional modification can be used to strike a better balance between accuracy and fairness at the client level.

**Summary of findings and contributions.**

*1. Unintended fairness benefits of personalization.* We conduct extensive numerical experiments, under different notions of fairness, and using both real-world and synthetic data, to show that personalization in FL (which is intended to improve local accuracy) can also improve local fairness as an unintended benefit (Section 4). We highlight the potential statistical and computational reasons leading to this alignment, and provide analytical support under certain conditions (Prop. 1 and 2).

*2. New fairness-aware, personalized federated clustering algorithms.* We propose two new algorithms, Fair-FCA and Fair-FL+HC, by modifying the existing IFCA (Ghosh et al., 2020) and FL+HC (Briggs et al., 2020) personalized FL algorithms, to take fairness into account when (iteratively) determining clients' cluster memberships (Algorithms 1 and 2). We show that this can provide desired fairness-accuracy trade-offs by adjusting a hyperparameter (Section 5), and elaborate on when/how this technique can be applied to other (clustering-based) personalized FL methods.

## 2 RELATED WORK

Our work is at the intersection of two literatures: personalized FL, and achieving fairness in FL. Here, we discuss the most closely related works; we review additional related work in Appendix A.

**Personalized FL.** Existing literature can be categorized based on how personalization is achieved. Here, we investigate personalization through *clustering*, *local fine-tuning*, and *regularization*.
*Clustering:* Mansour et al. (2020) use a hypothesis-based clustering approach by minimizing the sum of loss over all clusters. Sattler et al. (2020) use the idea that cosine similarity between weight updates of different clients is highly indicative of the similarity of data distribution. Nardi et al. (2022) use a decentralized learning idea by exchanging local models with other clients to find the neighbor/group which has a high accuracy even using other clients' models. Zheng et al. (2022) learn a weighted and directed graph that indicates the relevance between clients. Ghosh et al. (2020) use a distributed learning idea by broadcasting all clients models to others, and collecting back the cluster identity from clients who can identify good performance when using others' models.
*Local fine-tuning:* Fallah et al. (2020) propose using a Model Agnostic Meta Learning (MAML) framework, where clients run additional local gradient steps to personalize the global model. Arivazhagan et al. (2019); Jiang & Lin (2022) propose using deep learning models with a combination of feature extraction layers (base) and global/local head (personalization). Jiang & Lin (2022), inspired by Arivazhagan et al. (2019), further consider robustifying against distribution shifts.
*Model regularization:* Hanzely & Richtárik (2020); Sahu et al. (2018); Li et al. (2020; 2021b) add a regularization term with mixing parameters to penalize the distance between the local and global models. In particular, Sahu et al. (2018) has a pre-set regularization parameter and allows for system heterogeneity. Li et al. (2021b) consider improving accuracy, while being robust to data and model poisoning attacks, and fair. Similarly, T Dinh et al. (2020) formulate a bi-level optimization problem, which helps decouple personalized model optimization from learning the global model. Huang et al. (2021) propose the FedAMP algorithm which also introduces an additional regularization term, but differs from previous works in that they encourage similar clients to collaborate more.

In this paper, we will consider three variants of clustered FL algorithms (`IFCA` (Ghosh et al., 2020), `FL+HC` (Briggs et al., 2020), and `CFL-Cosine` (Sattler et al., 2020)) from the clustering category. Additionally, we consider `MAML-FL` by Fallah et al. (2020) from the local fine-tuning category, and `FedProx` by Li et al. (2020) from the model regularization category. Our new proposed algorithms, `Fair-FCA` and `Fair-FL+HC`, will belong to the clustering category.

**Fairness in FL.** This literature, surveyed recently in Shi et al. (2023); Rafi et al. (2023), can also be categorized depending on the adopted notion of fairness. Notably, several works (e.g., (Li et al., 2019a; 2021b; Zhang et al., 2021; Wang et al., 2021)) on fair FL consider a notion of *performance fairness*, assessing whether the learned model(s) achieve uniform accuracy across all clients. In contrast, we focus on notions of *social (group) fairness* (also studied in, e.g., (Abay et al., 2020; Zhang et al., 2020; Du et al., 2021; Gálvez et al., 2021; Zeng et al., 2021; Ezzeldin et al., 2023)), which compare statistics such as selection rate, true positive rate, or false positive rate, across different demographic/protected groups, to identify any bias in how humans will be treated once the learned models are deployed. In contrast to all these works, we do not aim to impose a fairness constraint during training or aggregation of the (global) models, but show that improved (local) fairness (and a better fairness-accuracy tradeoff) can be achieved either by personalization alone, or by grouping clients into clusters based on a fairness-aware cluster assignment metric.

## 3 PROBLEM SETTING AND PRELIMINARIES

We study a federated learning setting consisting of $n$ clients. Each client $i$ is tasked with a binary classification problem. The client's dataset consists of samples $z = (x, y, g)$, where $x \in \mathbb{R}^n$ is a feature, $y \in \{0, 1\}$ is the true label of the sample, and $g \in \{a, b\}$ is a demographic group membership based on a protected attribute (e.g., race, gender, or age). Client $i$'s samples are drawn independently at random from joint feature-label-group distributions with pdfs $\{f_g^{y,i}(x), y \in \{0, 1\}, g \in \{a, b\}\}$; note that these distributions are client-dependent and potentially different, motivating the need for personalization of the learned models. Let $\mathcal{D}_i$ denote the local dataset of client $i$.

**Standalone learning.** Clients can use their local data $\mathcal{D}_i$ to (independently) learn binary classifiers $h_{\theta_i}(x) : \mathbb{R}^n \rightarrow \{0, 1\}$, where $\theta_i$ are the parameters of the classifier (e.g., weights of a NN).

**Federated learning.** While clients can learn standalone models, Federated Learning (FL) paradigms enable them to leverage the power of collaborative learning. This is exemplified by the `FedAvg` al-

gorithm (McMahan et al., 2017), in which the objective is to learn a single *global* model $\theta$ that minimizes the average loss across all clients. Learning happens over rounds. During each communication round $t$, each client $i$ receives the latest global model $\theta^{t-1}$ from the server, performs a local update on this model based on its local data $\mathcal{D}_i$, and sends back its updated model $\theta_i^t$ to the server. The server then aggregates all the received local models (through a weighted sum) to create the new global model for the next round. While FedAvg can successfully learn a global model, if the clients' datasets $\mathcal{D}_i$ are heterogeneous, the local loss of the shared global model may be high locally for each client. This has motivated the development of personalized FL algorithms, five of which we consider in this work: clustering-based methods (specifically, IFCA, FL+HC, and CFL-Cosine), a local fine-tuning method (MAML-FL), and one that imposes model regularization (FedProx).

**Personalized Federated learning.** We provide a high-level description of the personalized FL methods considered in this paper. The IFCA algorithm of Ghosh et al. (2020) alternates between two steps: clustering/grouping clients based on model performance, and optimizing model parameters via gradient descent in each cluster separately, leading to distinct, *cluster-specific models*. The FL+HC algorithm of Briggs et al. (2020) uses a hierarchical clustering technique to group clients while minimizing intra-cluster variance, measured by the Euclidean distance of their models. The CFL-Cosine algorithm of Sattler et al. (2020) bipartitions clients, aiming to minimize the maximum cosine similarity of clients' gradient updates. The MAML-FL algorithm of Fallah et al. (2020) builds upon the FedAvg approach by allowing clients to further *fine-tune* the received global model through additional local gradient steps, thereby specializing it to their data and reducing local loss. Similarly, the FedProx algorithm of Li et al. (2020) introduces a $l_2$ regularization term for each client to learn a mix of local and global models.

**Assessing Fairness.** Given the sensitive demographic information $g$ (e.g., race, gender, age) contained in each data point in our problem setting, we further assess the *group fairness* of the learned models. Different notions of group fairness have been proposed, each assessing a specific statistical disparity in how the model treats individuals from different groups. Here, we consider three notions of fairness: (1) *Statistical Parity* (SP) (Dwork et al., 2012; Ezzeldin et al., 2023), which compares the difference in selection rate between the two groups (i.e., $\Delta_{\text{SP}}(\theta) := |Pr(\hat{y}(\theta) = 1|g = a) - Pr(\hat{y}(\theta) = 1|g = b)|$, where $\hat{y}(\theta)$ is the label assigned to data points under model $\theta$); (2) *Equality of Opportunity* (EqOp) (Hardt et al., 2016), which compares the differences in true positive rates across groups (i.e., $\Delta_{\text{EqOp}}(\theta) := |Pr(\hat{y}(\theta) = 1|g = a, y = 1) - Pr(\hat{y}(\theta) = 1|g = b, y = 1)|$); and (3) *Equalized Odd* (EO) (Hardt et al., 2016), which is set to either the true positive rate or false positive rate difference between the two protected groups, whichever larger (i.e., $\Delta_{\text{EO}}(\theta) := \max_{i \in \{0,1\}} |Pr(\hat{y}(\theta) = 1|g = a, y = i) - Pr(\hat{y}(\theta) = 1|g = b, y = i)|$).

**Comparing different algorithms.** To compare the different algorithms' performance, we will contrast the (average of the) *local accuracy* of each client's local model $\theta_i$ (which can be obtained by finding its standalone model, the global FedAvg model, a cluster-specific local model, or a fine-tuned local model, or a regularized local model) when applied to the client's local data $\mathcal{D}_i$. We will similarly contrast the different algorithms in terms of the (average of) *local $f$-fairness*, evaluating each client's local model on its local data under $f \in \{\text{SP}, \text{EqOp}, \text{EO}\}$.

## 4 PERSONALIZATION CAN ALSO IMPROVE FAIRNESS

We begin by numerically illustrating the (unintended) fairness benefits of personalization, and elaborate on the reasons behind it. (We substantiate these with analytical support in Section B.) We will compare the average local statistical parity (SP) fairness achieved by personalized FL algorithms, IFCA (Ghosh et al., 2020), FL+HC (Briggs et al., 2020), CFL-Cosine (Sattler et al., 2020), MAML-FL (Fallah et al., 2020), and FedProx (Li et al., 2020), against FedAvg (McMahan et al., 2017) and Standalone learning. Similar experiments supporting our findings under other notions of fairness (EqOp, EO) are given in Appendix C.2.

We first present numerical experiments on the "Retiring Adult" dataset (Ding et al., 2021). The dataset consists of census data collected from 50 U.S. states, and Puerto Rico. In our context, each state is a client within the FL framework. Each data sample includes multi-dimensional features $x$ (e.g., age, education, citizenship), a true label $y$, and a protected attribute $g$ (race, sex). To provide a more clear comparison of the effects of personalization, we have manually scaled the feature set ($x$) by 60% for the states with IDs $\{1, 10, 20, 30, 40, 50\}$; this exacerbates the data heterogeneity. We focus on two binary classification tasks: Employment (ACSEmployment) and Income (ACSIn-

come), and employ a two-layer neural network for both tasks. Each client is provided with 1000 training samples and 2000 testing samples. The values reported are the averages from 5 runs. Further details regarding the dataset and models, and additional figures, can be found in Appendix C.1. We also conduct similar numerical experiments under ACSHealth, another task in the Retiring Adult dataset, and find the same insights; these can be found in Appendix C.3. We will also present numerical experiments on the "Adult" dataset (Dua & Graff, 2017) in Section 4.3, and on synthetic data in Appendix D.

### 4.1 IMBALANCED GROUPS: STATISTICAL ADVANTAGES OF COLLABORATION

We first consider the ACSEmployment task with "race" as the protected attribute. Fig 1(a) shows the fraction of samples in each group-label, from several states, highlighting an imbalance between samples from the White and Non-White groups. This is further evident in Figure 1(b), which shows that most states have only $\sim 10\%$ qualified (label 1) samples from the Non-White group, in contrast to $\sim 35\%$ qualified samples from the White group.

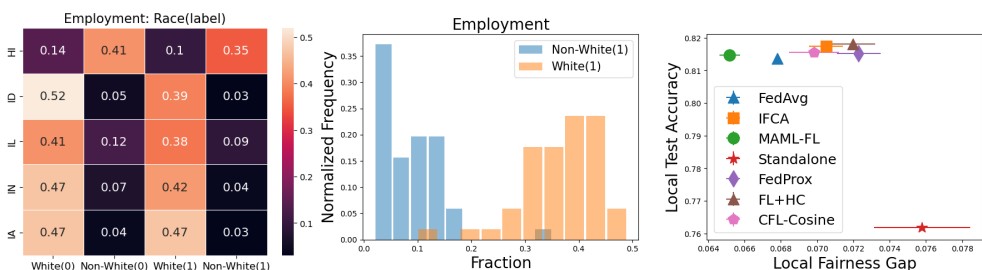

(a) Fraction of samples   (b) Normalized sample frequency   (c) Local accuracy vs. fairness gap

Figure 1: Experiments on the ACSEmployment task with imbalanced groups (race).

Fig 1(c) shows that all collaborative training algorithms (FedAvg, MAML-FL, IFCA, FedProx, FL+HC, and CFL-Cosine) achieve better local fairness (smaller gap) compared to Standalone learning. This is due to the *statistical benefits* of collaboration: each client has limited samples in the non-White group, leading to poorly trained models with high local fairness gap (and low accuracy). In contrast, collaborative training in essence has access to more data, improving both metrics. For the same reason, the IFCA, FL+HC and CFL-Cosine algorithms, which partition clients into multiple clusters, has (slightly) worse local fairness compared to FedAvg. Similarly, the FedProx algorithm imposes a regularization term that prevents the local updates from deviating too much from the global model, making it less fair compared to FedAvg. In comparison, the MAML-FL algorithm, which effectively sees the global dataset (when training the global model that is later fine-tuned by each client), has better local fairness compared to FedAvg, indicating that personalization can improve both local accuracy (as intended) and local fairness (as a side benefit).

### 4.2 BETTER-BALANCED GROUPS: COMPUTATIONAL ADVANTAGES OF COLLABORATION

We next consider better-balanced data, to show advantages of collaborative and personalized training beyond the statistical benefits of (effectively) expanding training data. We again consider the ACSEmployment task, but now with "sex" as the protected attribute. Fig 2(a) shows that data samples are more evenly distributed across groups and labels in this problem. Figure 2(b) further confirms that clients exhibit similar sample fractions of label 1 individuals in male and female groups.

We first notice from Fig 2(c) that all collaborative training algorithms still have better local fairness compared to Standalone learning. Furthermore, we observe that all personalized learning algorithms (IFCA, FL+HC, CFL-Cosine, MAML-FL, and FedProx) improve both local accuracy and local fairness compared to FedAvg. This is due to the *computational advantages* of (personalized) collaborative learning: for each client, due to similarity of the data for the male and female groups (as seen in Figure 2(b)) the objective of maximizing local accuracy is aligned with reducing the local fairness gap. Therefore, collaboration improves local fairness, with personalization further enhancing the model's local accuracy and therefore its fairness.

We also note that (local) accuracy and fairness may not necessarily be aligned. Our next experiment shows that personalization can still improve fairness in such tasks compared to non-personalized

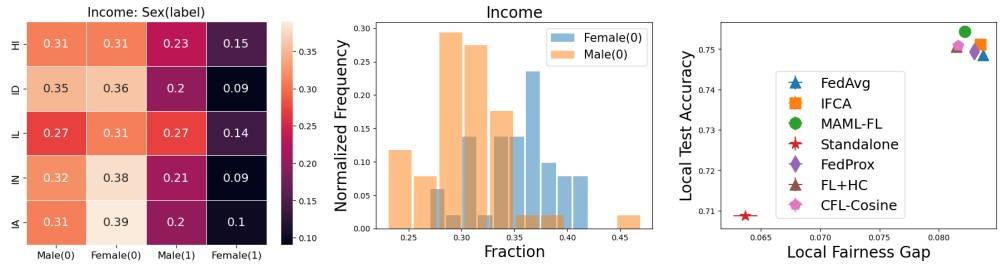

(a) Fraction of samples    (b) Normalized sample frequency    (c) Local accuracy vs. fairness gap

Figure 2: Experiments on the ACSEmployment task with better-balanced groups (sex).

`FedAvg`, which (we interpret) is driven by a combination of statistical and computational benefits. Specifically, we conduct experiments on another task, ACSIncome, with "sex" as the protected attribute. Fig 3(a) shows that for this task, the fraction of samples is comparable across groups for label 0 data, but differs for label 1 data. From Fig 3(c), we observe that this time, all collaborative training algorithms improve accuracy but have *worse* local fairness compared to `Standalone` learning; this is because improving (local) accuracy is not aligned with fairness in this task. That said, we observe that the personalized FL algorithms slightly improve local fairness compared to `FedAvg`. We interpret this as the statistical advantage of (effectively) observing more label 1 data (as `FedAvg` does, too), combined with a computational advantage of not overfitting a global model to the majority label 0 data (unlike what `FedAvg` may be doing).

(a) Fraction of samples    (b) Normalized sample frequency    (c) Local accuracy vs. fairness gap

Figure 3: Experiments on the ACSIncome task with sex as the protected attribute.

### 4.3 EXPERIMENTS ON THE ADULT DATASET

The `Adult` dataset (Dua & Graff, 2017) is a widely used dataset from the UCI Machine Learning Repository, containing demographic data from the 1994 U.S. Census. Its task is to predict whether an individual's income exceeds $50k annually, based on a combination of features on the individual, including their demographics. The dataset consists of 48,842 samples described by 14 attributes, both categorical and numerical. Among these, 41,762 samples belong to the White group, while 7,080 samples are from the Non-White groups. Given the nature of this data's heterogeneity, we randomly generate 5 clients each with an unbalanced number of samples based on race. Additionally, in Appendix C.3, we conduct experiments where samples are distributed with less heterogeneity across 5 clients, as done in other existing FL studies (e.g. (Ezzeldin et al., 2023)).

| Client ID | Non-White samples | White samples |
|---|---|---|
| 0 | 300 | 1000 |
| 1 | 300 | 1000 |
| 2 | 300 | 39362 |
| 3 | 600 | 200 |
| 4 | 5580 | 200 |

(a) Number of samples    (b) Local accuracy vs. fairness gap

Figure 4: Experiments on the `Adult` dataset with race as the protected attribute.

From Fig 4(b), we can see that the results are consistent with our findings in Section 4.1. Interestingly, we observe that the `IFCA` algorithm clusters clients by grouping those with more White samples into one cluster and those with more Non-White samples into another (i.e., {0,1,2}, {3,4}). In contrast, the `FL+HC` clusters clients by grouping those with more samples into one cluster and those with less samples into another (i.e., {0,1,3,4}, {2}). As a result, these two variants of clustering-based algorithms have different performance, having statistical advantages but for different reasons.

### 4.4 COMPARISON WITH A FAIR FEDERATED LEARNING ALGORITHM

A natural question that may arise is why not utilize an existing *fair* FL algorithm to improve fairness, as these might offer superior fairness compared to a personalized FL algorithm. Indeed, if one only focuses on improving (global) fairness, choosing a dedicated fair FL algorithm could be the best choice. However, here we point to the additional (cost-free) local fairness improvements achieved through the incorporation of personalization in FL. Our findings also suggest that this leads to *a more favorable trade-off* between fairness and accuracy, inspiring our new algorithm in Section 5.

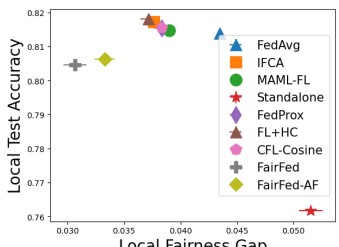

Figure 5: Comparing `FedAvg`, `FairFed`, `IFCA`, `MAML-FL`, `FedProx`, `FL+HC`, `CFL-Cosine` and `Standalone` learning.

To show this, we consider the FairFed algorithm of Ezzeldin et al. (2023), which adjusts the global model aggregation weights according to a fairness metric with the goal of improving (global) fairness. We also consider an extension of the algorithm, FairFed-AF, which adjusts the weights in aggregation according to both accuracy and fairness metrics. The experiments in Fig. 5 are on the ACSEmployment task with sex as the protected attribute. We observe that fair FL algorithms can achieve the best fairness across all algorithms. However, they have worse local accuracy compared to other collaborative training algorithms, as they focus (at least partially) on improving fairness and not accuracy.

### 4.5 ANALYTICAL SUPPORT

To support and validate our findings from the numerical experiments, we also analytically show that personalized Federated clustering algorithms (which cluster/group similar clients to improve their models' local accuracy) can also lead to better local fairness, when compared to a (non-personalized) shared global model, under certain conditions. Our results are detailed in Appendix B. Specifically, we consider a setting where data has single-dimensional, normally distributed features, leading to optimal threshold-based classifiers, and assume that clients can be put into two clusters $C_\alpha$ and $C_\beta$ based on similarities in their local datasets $\mathcal{D}_i$.

We start with the `EqOp` (Equality of Opportunity) fairness constraint, which aims to equalize true positive rates (TPR) between the protected groups $a$ and $b$. Our first result (Proposition 1) shows that if $\theta_\alpha^* < \theta_\beta^*$ (i.e., the data heterogeneity is such that cluster $C_\alpha$ has a lower optimal threshold than $C_\beta$), then clients in cluster $C_\alpha$ can obtain better local fairness (in addition to better local accuracy) with their cluster-specific model compared to if they used a global model shared with clients in $C_\beta$.

Our second result (Proposition 2) considers the `SP` (statistical parity) constraint, which assesses the disparity in the selection (positive classification) rate between the two protected groups. This is impacted by both the group $a$ vs. $b$ feature distributions *as well as* the label rates, rendering it more stringent than `EqOp` fairness. Figure 6 illustrates this by plotting the fairness gap vs. the decision threshold $\theta$ for `SP` vs. `EqOp`, showing that `SP` exhibits less struc-

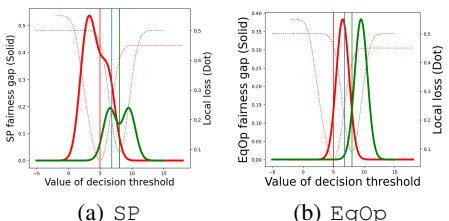

(a) `SP`       (b) `EqOp`

Figure 6: Fairness gap vs $\theta$.

tured changes as the decision threshold moves (e.g., due to the use of a global model). Therefore, to facilitate theoretical analysis, we impose additional assumptions on the data distributions. Intuitively, the resulting proposition states the following: when there are more label 1 samples in both groups, the global model $\theta_G^*$ will pull the $C_\alpha$ cluster model $\theta_\alpha^*$ up to account for the label imbalance, resulting in a deterioration in both fairness and accuracy for clients in this cluster. Similarly, if group $a$ ($b$)'s clients are majority label 1 (0), then, the use of global instead of a local models can result in a higher fairness gap for clients in $C_\alpha$ for the same reason as earlier.

## 5 FAIRNESS-AWARE FEDERATED CLUSTERING ALGORITHMS

We have observed numerically (Section 4) and analytically (Appendix B) that integrating personalization techniques (such as clustering) in the FL paradigm can provide a dual benefit by not only enhancing local accuracy as intended, but also improving (local) group fairness. Motivated by this, we propose to further enhance the fairness of the learned personalized models by modify the clustering assignment process to also include fairness considerations.

We illustrate this idea by proposing new fairness-aware Federated clustering algorithms, `Fair-FCA` and `Fair-FL+HC`, based on the existing `IFCA` algorithm (Ghosh et al., 2020) and `FL+HC` algorithm (Briggs et al., 2020), respectively. We choose these algorithms as many existing algorithms in the clustered FL literature are built on the `IFCA` framework (e.g., Li et al. (2021a); Chung et al. (2022); Huang et al. (2023); Ma et al. (2024)) and the `FL+HC` framework (e.g., Jothimurugesan et al. (2023); Luo et al. (2023); Li et al. (2023); Sun et al. (2024).

### 5.1 INTEGRATING FAIRNESS METRICS IN CLUSTER IDENTITY ASSIGNMENT

Consider a FL scenario involving a total of $n$ clients and $K$ disjoint clusters. Each client $i$ has $n_i$ samples $Z_i := \{z_{ij}\}_{j=1}^{n_i}$. Let $f(z, \theta)$ be the loss function associated with data point $z$ under model $\theta$. Then, the empirical loss of client $i$ is given by $F(Z_i, \theta) := \frac{1}{n_i} \sum_j f(z_{ij}, \theta)$. Also, let $\Psi^f(Z_i, \theta)$ denote the fairness under model $\theta$ for fairness metric $f \in \{\text{SP}, \text{EqOp}, \text{EO}\}$ assessed on $Z_i$.

**The `Fair-FCA` algorithm.** This algorithm iterates over two steps: (1) cluster identity assignment, and (2) training of cluster-specific models. Specifically, let $\Theta_k^t$ denote cluster $k$'s model at time step $t$. The cluster identity for client $i$ at time $t$, denoted $c^t(i)$, is determined by:

$$c^t(i) = \arg \min_{k \in [K]} \gamma F(Z_i, \Theta_k^t) + (1 - \gamma) \Psi^f(Z_i, \Theta_k^t) \tag{1}$$

Here, $\gamma$ is a hyperparameter that strikes a desired balance between accuracy and fairness. For $\gamma = 1$, we recover the `IFCA` algorithm; for $\gamma = 0$, we obtain a clustered FL algorithm that prioritizes (local) $f$-fairness only when grouping clients. For $0 < \gamma < 1$, we obtain clusters that provide each client with the best fairness-accuracy tradeoff among those attainable if the client were to join each cluster. Let $C_k^t$ be the set of clients whose cluster identity is $k$ at the end of this assignment process (i.e., $C_k^t = \{i \in [n] : c^t(i) = k\}$).

Once clients get assigned clusters, each client $i$ starts from its corresponding cluster model $\Theta_{c^t(i)}^t$, and locally runs gradient steps to update this model, such that $\theta_i^t = \Theta_{c^t(i)}^t - \eta \nabla_{\theta_i} F(Z_i, \Theta_{c^t(i)}^t)$. Then, the updated local models $\theta_i^t$ are sent to the central server, who uses these to update the cluster models to $\Theta_{1:K}^{t+1}$ by taking the weighted average of the local models of clients in corresponding clusters. Formally, $\Theta_k^{t+1} = \Theta_k^t - \sum_{i \in C_k^t} \frac{n_i}{\sum_i n_i} (\Theta_k^t - \theta_i^t), \forall k \in [K]$. The pseudo-code for `Fair-FCA` is shown in Algorithm 1.

**The `Fair-FL+HC` algorithm.** Initially, the algorithm runs the regular `FedAvg` procedure for a predetermined number of rounds before clustering. Once a global model $\theta^{FA}$ is obtained, each client receives the model and performs several local updates to personalize their local models $\theta_i$.

Like the `FL+HC` algorithm, the `Fair-FL+HC` also employs hierarchical clustering with a set of hyperparameters $P$ to group clients by minimizing intra-cluster variance, measured using the $L_2$ Euclidean distance metric. We extend this approach by also considering fairness performance.

$$\text{Clusters} = \text{HierarchicalClustering}(\gamma \mathbf{D} + (1 - \gamma) \mathbf{\Psi}^f(Z, \theta), P) \tag{2}$$

Here, $\mathbf{D}$ is a symmetric matrix where each entry $D_{i,j}$ represents the Euclidean distance between $\theta_i$ and $\theta_j$. $\mathbf{\Psi}^f(Z, \theta) := max(\Psi^f(Z_i, \theta_j), \Psi^f(Z_j, \theta_i))$, with $f \in \{\text{SP}, \text{EqOp}, \text{EO}\}$ is also a symmetric matrix that captures the worst-case $f$-fairness performance when client $i$'s model is evaluated on client $j$'s local data, or vice versa. The parameter $\gamma$ balances between fairness and accuracy considerations; when $\gamma = 1$, we recover the `FL+HC` algorithm. Once clustering is completed, each cluster trains its model independently using `FedAvg`. The pseudo-code for `Fair-FL+HC` is shown in Algorithm 2.

## 5.2 Fairness-accuracy tradeoff using Fair-FCA and Fair-FL+HC

We begin by conducting a numerical experiment on a synthetic dataset, to illustrate the ability of Fair-FCA and Fair-FL+HC to strike a balance between fairness and accuracy. We consider a total of 8 clients that could (potentially) be clustered into two clusters. Among these, 6 clients (Client ID: 2,4,5,6,7,8) have similar data distributions, with 4 clients (Client ID: 4,6,7,8) sharing identical distributions across the two protected groups $a, b$ (low fairness gap). The remaining 2 clients (Client ID: 1,3) have different data distributions compared to the first six, but they also share identical distributions across the two protected groups. We consider $f = \text{SP}$. Let

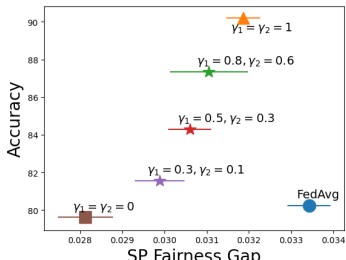

Figure 7: Acc. vs. SP-fairness

$\gamma_1, \gamma_2 \in [0, 1]$ be the hyperparameters of the Fair-FCA and Fair-FL+HC algorithms, respectively. Additionally, let the distance threshold $d \in P$ in Fair-FL+HC be 0.35.

Figure 7 shows that when $\gamma_1 = \gamma_2 = 1$, both Fair-FCA and Fair-FL+HC prioritizes accuracy; by design, this is attained by grouping the 6 clients having similar data distributions together ({1,3} and {2,4,5,6,7,8}). Similarly, when $\gamma_1 = \gamma_2 = 0$, both Fair-FCA and Fair-FL+HC focus only on SP fairness, this time clustering clients that have identical distributions on the two protected groups together ({2,5} and {1,3,4,6,7,8}). Lastly, by setting $\gamma_1, \gamma_2 \in (0, 1)$, we can effectively account for both accuracy and SP fairness when clustering: when $\gamma_1 = 0.3, \gamma_2 = 0.1$, the clusters are {2,4,5} and {1,3,6,7,8}; when $\gamma_1 = 0.5, \gamma_2 = 0.3$, the clusters are {2,4,5,6} and {1,3,7,8}; and when $\gamma_1 = 0.8, , \gamma_2 = 0.6$, the clusters are {2,4,5,6,7} and {1,3,8}.

## 5.3 Comparing Fair-FCA vs. IFCA; Fair-FL+HC vs. FL+HC

We now compare the IFCA and FL+HC algorithms of Ghosh et al. (2020) and Briggs et al. (2020), respectively, and our Fair-FCA and Fair-FL+HC algorithms with $\gamma = 0.5$ and with $f = \text{SP}$, in terms of local accuracy and local group fairness. We proceed with the same experiment setting as in Section 4. (We also provide experiment results using the original Retiring Adult dataset without feature scaling in Appendix C.6, and experiments with EqOp as the fairness metric in Appendix C.4.)

Figure 8 verifies that under different datasets (ACSEmployment, ACSIncome) and different protected attributes (race, sex), Fair-FCA and Fair-FL+HC (symbols in squares and triangles) exhibits a fairness-accuracy tradeoff (improved fairness performance at the expense of accuracy degradation) consistent with the choice of $\gamma = 0.5$ compared to the IFCA and FL+HC algorithm.

Moreover, we can observe that FL+HC could achieve better accuracy compared to IFCA in the ACSEmployment tasks with both race and sex attributes. This is because IFCA uses a predetermined number of clusters, set to 2 in all experiments. As a result, it effectively groups scaled clients into one cluster and unscaled ones into the other. In contrast, FL+HC employs a hierarchical clustering technique, which does not require prior knowledge of the number of clusters. This allows FL+HC not only to successfully group scaled clients into one cluster but also to form additional clusters (if needed) for other different clients. The fairness performance of IFCA and FL+HC remains consistent with our findings in Section 4.

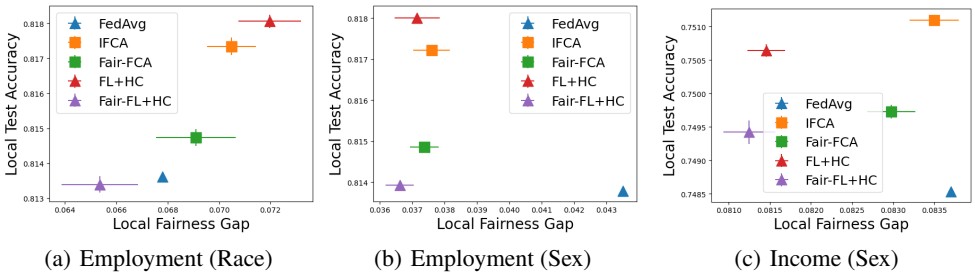

(a) Employment (Race)      (b) Employment (Sex)      (c) Income (Sex)

Figure 8: Fair-FCA and Fair-FL+HC with different tasks and protected attributes ($f = \text{SP}$)

---

**Algorithm 1:** `Fair-FCA`

---

**Input**: Number of clusters $K$, number of clients $n$, number of local updates $E$, cluster model initialization $\Theta_{1:K}$, learning rate $\eta$, fairness-accuracy tradeoff $\gamma$, fairness $f \in \{\texttt{SP}, \texttt{EqOp}, \texttt{EO}\}$.

**Initialize**: Start clusters $k \in [K]$ by randomly selecting one client for each.

**while** *not converge* **do**
    **for** *client $i \in [n]$* **do**
        **Find** cluster identity:
            $c(i) = \arg\min_{k \in [K]} \gamma F(Z_i, \Theta_k) + (1 - \gamma)\Psi^f(Z_i, \Theta_k)$
        **Initialize** $\theta_i = \Theta_{c(i)}$
        **Perform** $E$ steps of local update
            $\theta_i = \theta_i - \eta\nabla_{\theta_i}F(Z_i, \theta_i)$
        **Upload** $\theta_i$ to server
    **end**
    **Update** the cluster model $\Theta_{1:K}$
        $\Theta_k = \Theta_k - \sum_{i \in C_k} \frac{n_i}{\sum_i n_i}(\Theta_k - \theta_i)$
    **Send** new cluster models $\Theta_{1:K}$ to all clients
**end**

**Output**: Cluster models $\Theta_{1:K}$, Cluster identity $c(i), \forall i \in [n]$

---

**Algorithm 2:** `Fair-FL+HC`

---

**Input**: Number of rounds $k_1/k_2$ before/after clustering, number of clients $n$, number of local updates $E$, learning rate $\eta$, Model initialization $\theta$, Set of hyperparameters for the hierarchical clustering algorithm $P$, fairness-accuracy tradeoff $\gamma$, fairness $f \in \{\texttt{SP}, \texttt{EqOp}, \texttt{EO}\}$.

**for** *each round $t \in [k_1]$* **do**
    **Output** `FedAvg` model: $\theta = \texttt{FedAvg}(\theta, n)$
**end**
**for** *each client $i \in [n]$ in parallel* **do**
    **Initialize** $\theta_i = \theta$
    **Perform** $E$ steps of local update: $\theta_i = \theta_i - \eta\nabla_{\theta_i}F(Z_i, \theta_i)$
**end**
Clusters = HierarchicalClustering$(\gamma\mathbf{D} + (1 - \gamma)\mathbf{\Psi}^f(Z, \theta), P)$
**for** *each cluster $c \in Cluster$ in parallel* **do**
    **for** *each client $i \in c$ in parallel* **do**
        **Perform** `FedAvg` procedures: $\texttt{FedAvg}(\theta_i, n_c), n_c \in c$
    **end**
**end**

---

## 6 CONCLUSION

We have shown, both numerically and analytically, that there can be (unintended) fairness benefits to personalization in Federated Learning. We have identified potential statistical reasons (improved data diversity) and computational reasons (alignments between local accuracy and fairness while preventing overfitting) for this alignment. Motivated by these findings, we further proposed new fairness-aware Federated clustering algorithms, `Fair-FCA` and `Fair-FL+HC`, which take both the fairness and model performance into account when clustering clients. We find that this modification can lead to an improved fairness-accuracy tradeoff, (partly) as personalization technique offer dual benefits in terms of accuracy and fairness. Extending our analytical findings (both for clustered FL algorithms, and to other classes of personalized FL methods), and identifying methods to integrate fairness considerations into other (non-clustering based) personalized FL algorithms, are main directions of future work.

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
