# OpenReview forum: "Enhancing Group Fairness in Federated Learning through Personalization"
_ICLR.cc/2025/Conference — Submitted to ICLR 2025_

### Official Review · Reviewer_yfVM · 2024-10-31

**Soundness:** 3
**Presentation:** 2
**Contribution:** 2
**Rating:** 5
**Confidence:** 3

**Summary:**

In this paper, the authors investigate the impact of such personalization techniques on the group fairness of the learned models, and show that personalization can also lead to improved (local) fairness as an unintended benefit. The authors propose Fair-FCA and Fair-FL+HC algorithms which achieve state-of-the-art performance.

**Strengths:**

+ The authors investigate personalization to improve fairness which is somewhat interesting.

+ The authors conduct extensive experiments to validate their claims.

**Weaknesses:**

+ It is still unclear to me why personalization can also improve fairness. After reading through the paper, it seems that clustering could somewhat improve fairness.

+ The mathematical proof and illustrations should appear in the main paper since they are relatively important.

+ Some similar works should be discussed, e.g., [1]

Ref:

[1] Intra-and Inter-group Optimal Transport for User-Oriented Fairness in Recommender Systems

**Questions:**

NA

---

> ### Author Response · Authors · 2024-11-22
> **Response**
>
> Thank you for your valuable comments and suggestions.
>
> **Comment 1:** Thank you for your thorough review. As you observed, our experiments demonstrate that the personalized FL algorithm, originally designed to enhance accuracy, can also yield unintended fairness benefits under specific scenarios when compared to classical FedAvg algorithms. On the theoretical side, we focus on the clustered FL algorithm and provide conditions under which these unintended fairness benefits arise.
>
> **Comment 2:** Thank you for your valuable suggestion regarding the theoretical analysis. We agree with your feedback and will incorporate the analytical results into the main text in the revised draft.
>
> **Comment 3:** Thank you for the suggested reference. The referenced paper explores User-Oriented Fairness (UOF) in recommender systems, addressing disparities in recommendation quality between advantaged users (those with more satisfied recommendation results) and disadvantaged user groups. While both our work and the referenced paper propose new evaluation metrics that balance fairness and accuracy, our approaches differ in several key aspects: fairness settings (group fairness vs. performance fairness), problem contexts (federated learning vs. recommender systems), and the motivations for introducing these metrics.

---

### Official Review · Reviewer_ZVVo · 2024-10-31

**Soundness:** 2
**Presentation:** 3
**Contribution:** 2
**Rating:** 5
**Confidence:** 5

**Summary:**

This manuscript explores the intersection of personalized Federated Learning (FL) and group fairness. It effectively demonstrates that personalization techniques, which are typically employed to enhance model accuracy on local data, can also inadvertently improve fairness. The paper substantiates these claims through comprehensive numerical experiments comparing various FL algorithms and introduces novel fairness-aware federated clustering algorithms. These algorithms, Fair-FCA and Fair-FL+HC, extend existing IFCA and FL+HC frameworks to incorporate a fairness metric into the cluster-selection process, aiming to optimize both fairness and accuracy.

**Strengths:**

1．	The manuscript is well-organized, clearly presenting the methodology and findings.

2．	The manuscript offers a comprehensive experimental analysis across various algorithms, fairness notions, and datasets.

**Weaknesses:**

1.	The authors demonstrate that personalized federated learning (FL) improves fairness through experiments and intuitive analysis; however, the manuscript lacks theoretical justification for these findings.

2.	The experimental methods used to assess the impact of personalization on fairness are outdated and do not incorporate relevant studies from the past three years.

3.	The proposed approach merely adds a fairness-related loss to existing FL methods, offering insufficient innovation to significantly advance fairness in federated learning.

**Questions:**

See weakness.

---

> ### Author Response · Authors · 2024-11-22
> **Response**
>
> Thank you for your valuable comments and suggestions.
>
> **Comment 1:** Thank you for your careful reading. Due to space constraints, we have included our analytical results in Appendix B. Specifically, we focus on the clustered FL algorithm and provide conditions under which these unintended fairness benefits arise.
>
> **Comment 2:** Thank you for your comments regarding the currency of the methods. We selected five personalized FL algorithms from three key categories: local fine-tuning, regularization, and clustering. Although all the chosen algorithms were proposed in 2020, they remain widely recognized and serve as foundational approaches within their respective categories. More importantly, our focus is on the impact of personalization on local group fairness, which lies at the intersection of personalized FL and fair FL research. To the best of our knowledge, this work is the first to identify the (unintended) fairness benefits of personalization in FL. Existing studies predominantly concentrate on either improving local accuracy (personalized FL literature) or enhancing global fairness (fair FL literature), whereas our work bridges this gap.
>
> **Comment 3:** Thank you for your comments regarding the proposed algorithms. As partially addressed in Comment 2, our work primarily focuses on the unintended fairness benefit introduced by personalization in FL. Importantly, for the original personalized FL algorithms, we do not impose any fairness constraints or modify the weighted aggregation step. As such, our goal is not to develop an algorithm that achieves optimal or comparable (local) fairness relative to existing FL methods designed for local fairness. Instead, we compared several personalized FL algorithms to evaluate their fairness performance. Upon observing the unintended fairness benefit, we explored incorporating fairness into personalized FL and proposed corresponding fairness-aware FL algorithms (e.g., Fair-FCA, Fair-FL+HC). While much of the FairFL literature focuses on enhancing global fairness, research on improving local fairness in FL remains relatively limited. Existing approaches for improving local fairness either impose fairness constraints locally [1,2] or explore other types of fairness [3]. However, these methods often experience a local fairness-accuracy trade-off compared to the FedAvg approach, sacrificing accuracy to improve fairness. We agree that our proposed approach simply adds a fairness-related loss to existing FL methods. However, the key insight we aim to convey is that we can leverage the unintended fairness benefit to propose algorithms for a better fairness-accuracy trade-off. Through a straightforward modification, we demonstrate how this approach enables firms to tune the hyperparameter to strike a customizable balance between fairness and accuracy, allowing them to make decisions aligned with their specific priorities. Additionally, both Fair-FCA and Fair-FL+HC are grounded in personalization from the clustering category. Based on our observations, it would be an interesting direction for future work to explore whether similar unintended fairness benefits arise in other personalization categories, such as local fine-tuning and regularization.
>
> [1] Du, Wei, et al. "Fairness-aware agnostic federated learning." Proceedings of the 2021 SIAM International Conference on Data Mining (SDM). Society for Industrial and Applied Mathematics, 2021.
>
> [2] Meerza, Syed Irfan Ali, et al. "GLOCALFAIR: Jointly Improving Global and Local Group Fairness in Federated Learning." arXiv preprint arXiv:2401.03562 (2024).
>
> [3] Nafea, Mohamed, Eugine Shin, and Aylin Yener. "Proportional fair clustered federated learning." 2022 IEEE International Symposium on Information Theory (ISIT). IEEE, 2022.

---

> > ### Comment · Reviewer_ZVVo · 2024-11-23
> >
> > Thank you for the response. I will keep my rating unchanged.

---

### Official Review · Reviewer_GmZ5 · 2024-11-01

**Soundness:** 2
**Presentation:** 3
**Contribution:** 1
**Rating:** 5
**Confidence:** 3

**Summary:**

This paper studies the problem of group fairness in federated learning. The authors find that personalized federated learning unintentionally benefits group fairness. The authors further introduce a fairness metric into clustering to improve the trade-off between fairness and accuracy. Experiments were conducted on real-world datasets.

**Strengths:**

1.	The idea of unintentionally benefiting group fairness through personalization is indeed interesting.
2.	The paper is generally well-written, with a clear structure that is easy to follow.

**Weaknesses:**

1.	Since personalization can reduce the impact of dominant clients, the finding that personalization benefits group fairness is intuitive and not particularly surprising. Also, the results presented in the paper do not consistently support this finding. For instance, in Figure 3, federated methods improve the accuracy while increasing the fairness gap, which is inconsistent with Figures 1 and 2.
2.	The technical contribution of this paper is limited. The first introduced metric is merely a linear combination of fairness terms with a hyperparameter, making it unconvincing to claim that it "improves the fairness-accuracy trade-off" through hyperparameter tuning. How to determine the value of hyperparameters in practice? The second introduced metric is an incremental combination of hierarchical clustering.

**Questions:**

Please refer to weaknesses.

---

> ### Author Response · Authors · 2024-11-22
> **Response**
>
> Thank you for your valuable comments and suggestions.
>
> **Comment 1:** Thank you for your careful reading. As you noted, the personalization technique does not always provide fairness improvement as an unintended benefit. For example, in Fig. 3 the ACSIncome dataset with sex as a protected attribute, there are fewer samples in the female group with label 1. As a result, improving accuracy will lean the decision toward the majority of samples, which will sacrifice the fairness performance. Therefore, we could see that all collaborative learning algorithms improve accuracy but have worse fairness performance compared to the standalone training. As you noted, the personalization, which reduces the impact of domain clients, slightly improves local fairness compared to FedAvg. We interpret this as the combined effects between statistical advantage (effectively observing more label 1 data as FedAvg does, too) and computational advantage (not overfitting a global model
> to the majority label 0 data unlike what FedAvg may be doing).
>
> **Comment 2:** Thank you for your careful reading. Our work primarily focuses on the unintended fairness benefit introduced by personalization in FL. Importantly, for the original personalized FL algorithms, we do not impose any fairness constraints or modify the weighted aggregation step. As such, our goal is not to develop an algorithm that achieves optimal or comparable (local) fairness relative to existing FL methods designed for local fairness. Instead, we compared several personalized FL algorithms to evaluate their fairness performance. Upon observing the unintended fairness benefit, we explored incorporating fairness into personalized FL and proposed corresponding fairness-aware FL algorithms (e.g., Fair-FCA, Fair-FL+HC). While much of the FairFL literature focuses on enhancing global fairness, research on improving local fairness in FL remains relatively limited. Existing approaches for improving local fairness either impose fairness constraints locally [1,2] or explore other types of fairness [3]. However, these methods often experience a local fairness-accuracy trade-off compared to the FedAvg approach, sacrificing accuracy to improve fairness. We agree that our proposed approach simply adds a fairness-related loss to existing FL methods. However, the key insight we aim to convey is that we can leverage the unintended fairness benefit to propose algorithms for a better fairness-accuracy trade-off. Through a straightforward modification, we demonstrate how this approach enables firms to tune the hyperparameter to strike a customizable balance between fairness and accuracy, allowing them to make decisions aligned with their specific priorities. Additionally, both Fair-FCA and Fair-FL+HC are grounded in personalization from the clustering category. Based on our observations, it would be an interesting direction for future work to explore whether similar unintended fairness benefits arise in other personalization categories, such as local fine-tuning and regularization.
>
> [1] Du, Wei, et al. "Fairness-aware agnostic federated learning." Proceedings of the 2021 SIAM International Conference on Data Mining (SDM). Society for Industrial and Applied Mathematics, 2021.
>
> [2] Meerza, Syed Irfan Ali, et al. "GLOCALFAIR: Jointly Improving Global and Local Group Fairness in Federated Learning." arXiv preprint arXiv:2401.03562 (2024).
>
> [3] Nafea, Mohamed, Eugine Shin, and Aylin Yener. "Proportional fair clustered federated learning." 2022 IEEE International Symposium on Information Theory (ISIT). IEEE, 2022.

---

> ### Comment · Reviewer_GmZ5 · 2024-11-27
>
> Thank you for your response. I would like to maintain my rating.

---

### Official Review · Reviewer_pn33 · 2024-11-04

**Soundness:** 2
**Presentation:** 3
**Contribution:** 2
**Rating:** 3
**Confidence:** 4

**Summary:**

This paper explores the impact of personalization techniques on local fairness (i.e., the model’s fairness on each client’s local data) in federated learning. Extensive experiments compare the accuracy and fairness of personalized federated algorithms with FedAvg and standalone learning.  Results indicate that some personalized algorithms can improve model fairness while maintaining accuracy. Finally, this paper proposes fairness-aware Federated clustering algorithms based on existing methods to enforce local group fairness while enhancing accuracy.

**Strengths:**

1. The paper investigates fairness in FL from an interesting perspective, namely the effect of personalization techniques on local fairness.
2. The paper provides extensive numerical experiments comparing the accuracy and fairness of personalization methods with FedAvg across various scenarios (dataset, heterogeneity).
3. The experiments and methods are clear and easy to read.

**Weaknesses:**

1. Concerns about contributions. The paper spends a significant amount of space presenting experimental results that demonstrate the dual benefits of personalized federated approaches in accuracy and fairness, indicating that personalization is a promising research avenue for fair FL. However, the subsequent analytical support and the proposed approaches appear to lack significant contributions.
- Only federated clustering algorithms are analyzed, and the assumptions in the theoretical analysis are too strong for practical FL settings.
- The proposed methods build upon existing methods by incorporating an additional fairness performance metric. The detailed algorithmic steps (Algorithm 1,2) resemble those in prior studies Ghosh et al. (2020) and Briggs et al. (2020).
2. No effective local fair baseline in the experiments. The authors claim that the proposed method can enhance local accuracy while unintentionally improving local fairness. For empirical validation, the authors should compare the proposed methods against existing federated learning methods designed to improve local fairness. However, the experiments in Section 5 do not include FL methods specifically designed for local fairness.
3. The empirical analysis of the proposed methods may not be entirely convincing. Experiments in this paper are limited to comparisons. Additional experiments are required to validate its stability in different federated setting, e.g. heterogeneity, client numbers.
4. It is inappropriate to evaluate local fairness on methods specifically designed for global fairness (section 4.4), since previous work [1] has pointed out that global fairness differs from local fairness.
5. Including the theoretical analysis and more detailed experimental results of the proposed method in the main text, rather than in the appendix, would strengthen the paper.
[1] Hamman, Faisal, and Sanghamitra Dutta. Demystifying local & global fairness trade-offs in federated learning using partial information decomposition. ICLR, 2024.

**Questions:**

Beyond the above weak points, there are also additional questions:

What motivated the authors to utilize federated clustering algorithms to improve local fairness? Figure 1(C) indicates that, in highly heterogeneous data settings, these methods underperform FedAvg and MAML in the accuracy-fairness trade-off.

Given that fairness constraints are non-convex and non-differentiable, does incorporating fairness metrics in existing federated clustering algorithms pose potential convergence challenges?

---

> ### Author Response · Authors · 2024-11-22
> **Response**
>
> Thank you for your valuable comments and suggestions.
>
> **Comment 1:** Thank you for your careful reading. Our work primarily focuses on the unintended fairness benefit introduced by personalization in FL. Importantly, for the original personalized FL algorithms, we do not impose any fairness constraints or modify the weighted aggregation step. As such, our goal is not to develop an algorithm that achieves optimal or comparable (local) fairness relative to existing Fair FL methods. Instead, we compared several personalized FL algorithms to evaluate their fairness performance. Upon observing the unintended fairness benefit, we explored incorporating fairness into personalized FL and proposed corresponding fairness-aware FL algorithms (e.g., Fair-FCA, Fair-FL+HC). We agree that our proposed approach simply adds a fairness-related loss to existing FL methods. However, the key insight we aim to convey is that we can leverage the unintended fairness benefit to propose algorithms for a better fairness-accuracy trade-off. Through a straightforward modification, we demonstrate how this approach enables firms to tune the hyperparameter to strike a customizable balance between fairness and accuracy, allowing them to make decisions aligned with their specific priorities. Additionally, both Fair-FCA and Fair-FL+HC are grounded in personalization from the clustering category. Based on our observations, it would be an interesting direction for future work to explore whether similar unintended fairness benefits arise in other personalization categories, such as local fine-tuning and regularization. Furthermore, extending the theoretical analysis to other types of personalization techniques or to clustered FL under less restrictive assumptions is one of our ongoing works.
>
> **Comment 2:** Thank you for your careful reading. As partially addressed in Comment 2, our goal is not to develop an algorithm that achieves optimal or comparable local fairness relative to existing FL methods designed for local fairness. Indeed, if the sole objective is to improve local fairness, choosing a dedicated FL algorithm designed for local fairness could be the best choice. However, the key insight we aim to convey is the additional, cost-free local fairness improvements achieved through the incorporation of personalization in FL.
>
>
> **Comment 3:** Thank you for your suggestions regarding (network) heterogeneity. We agree that considering both data and network heterogeneity could yield more comprehensive and promising results in FL. However, our work primarily focuses on the impact of data heterogeneity on fairness within a distributed dataset.
>
> **Comment 4:** Thank you for your suggestion regarding the selection of methods. As we mentioned, the key insight we aim to convey is the additional, cost-free local fairness improvements achieved through the incorporation of personalization in FL. Following your comments, we recognize that the current evaluation may not provide the most appropriate comparison. We will revise the experiments in the revised draft.
>
> **Comment 5:** Thank you for your valuable suggestion regarding the theoretical analysis and experiment results. We agree with your feedback and will incorporate the analytical results into the main text in the revised draft.

---

> > ### Author Response · Authors · 2024-11-22
> > **(Cont.) Response**
> >
> > **Additional 1:** Thank you for your careful reading. As you noted, the underperformance stems from the fact that clustered FL approaches partition clients into multiple clusters, thereby accessing less information compared to FedAvg. Consequently, this leads to worse local fairness performance. Similarly, as you observed, the MAML algorithm, which belongs to the local fine-tuning personalization category, can improve local fairness compared to FedAvg. As we addressed in previous comments, our primary goal is not to achieve optimal or comparable fairness performance relative to existing algorithms but rather to emphasize the unintended, cost-free local fairness improvements enabled by personalization techniques. One of our motivations is that while much of the FairFL literature focuses on enhancing global fairness, research on improving local fairness in FL remains relatively limited. Existing approaches for improving local fairness either impose fairness constraints locally [1,2] or explore other types of fairness [3]. However, these methods often experience a local fairness-accuracy trade-off compared to the FedAvg approach, sacrificing accuracy to improve fairness. In contrast, we propose a local metric that balances both local accuracy and local fairness, demonstrating how the unintended fairness benefit can be leveraged to improve both simultaneously. Following your comments, we could explore integrating fairness into the MAML algorithm (e.g., Fair-MAML) to achieve a better fairness-accuracy trade-off. Extending the theoretical and numerical analysis to other personalization techniques (e.g., local fine-tuning, regularization, etc.) is an interesting direction for future work.
> >
> > **Additional 2:** Thank you for your comment. We agree that incorporating a fairness metric into the cluster identity assignment step introduces challenges in establishing convergence guarantees. Specifically, the combined objective, which now includes a linear combination of loss and fairness, becomes more complex. While the loss function is typically convex and differentiable, fairness metrics are often neither convex nor differentiable. Determining how to impose appropriate and not overly restrictive assumptions to derive meaningful convergence guarantees is one of our ongoing works.
> >
> > [1] Du, Wei, et al. "Fairness-aware agnostic federated learning." Proceedings of the 2021 SIAM International Conference on Data Mining (SDM). Society for Industrial and Applied Mathematics, 2021.
> >
> > [2] Meerza, Syed Irfan Ali, et al. "GLOCALFAIR: Jointly Improving Global and Local Group Fairness in Federated Learning." arXiv preprint arXiv:2401.03562 (2024).
> >
> > [3] Nafea, Mohamed, Eugine Shin, and Aylin Yener. "Proportional fair clustered federated learning." 2022 IEEE International Symposium on Information Theory (ISIT). IEEE, 2022.

---

### Meta-Review · Area_Chair_2kNP · 2024-12-21

**Metareview:**

The paper introduces two fairness-aware clustering algorithms, Fair-FCA and Fair-FL+HC that incoperate fairness metrics into clustering process. Reviewers appreciated the clarity and organization of the manuscript, as well as the comprehensive experimental analysis across datasets and scenarios.

However, several weaknesses were noted. While the findings regarding the fairness benefits of personalization are interesting, they are largely intuitive and lack theoretical justification. Additionally, the experiments do not include effective baselines for local fairness or incorporate recent advancements in fairness-focused FL methods. Reviewers also questioned the practical impact of the proposed algorithms, given strong assumptions. Overall, while the paper addresses an important problem, it falls short in both theoretical contributions and experimental evaluation, making it unsuitable for publication at this venue in its current form.

**Additional Comments On Reviewer Discussion:**

The original ratings for this paper were all negative. During the rebuttal phase, no reviewers changed the scores. I carefully read authors' responses; the authors acknowledge the some concerns but did not address them during the rebuttal.

---

### Decision · Program_Chairs · 2025-01-22

Reject